# Changes in Substance Use and Mental Health Burden among Women during the Second Wave of COVID-19 in Germany

**DOI:** 10.3390/ijerph18189728

**Published:** 2021-09-15

**Authors:** Henrike Schecke, Madeleine Fink, Alexander Bäuerle, Eva-Maria Skoda, Adam Schweda, Venja Musche, Hannah Dinse, Benjamin Maurice Weismüller, Sheila Moradian, Norbert Scherbaum, Martin Teufel

**Affiliations:** 1Department of Addictive Behavior and Addiction Medicine, Medical Faculty, LVR-Hospital Essen, University of Duisburg-Essen, Virchowstrasse 174, D-45147 Essen, Germany; norbert.scherbaum@uni-due.de; 2Clinic for Psychosomatic Medicine and Psychotherapy, LVR University Hospital Essen, University of Duisburg-Essen, Virchowstrasse 174, D-45147 Essen, Germany; madeleine.fink@lvr.de (M.F.); Alexander.Baeuerle@uni-due.de (A.B.); eva-maria.skoda@uni-due.de (E.-M.S.); adam.schweda@lvr.de (A.S.); venja.musche@lvr.de (V.M.); hannah.dinse@lvr.de (H.D.); benjaminmaurice.weismueller@lvr.de (B.M.W.); sheila.moradian@lvr.de (S.M.); martin.teufel@lvr.de (M.T.)

**Keywords:** COVID-19, substance use, alcohol, cannabis, mental health, depression

## Abstract

Unlike men, who are disproportionately affected by severe disease progression and mortality from COVID-19, women may be more affected by the economic, social and psychological consequences of the pandemic. Psychological distress and mental health problems are general risk factors for increases in the use of alcohol and other substances as a dysfunctional coping mechanism. Methods: An analysis was carried out of the female subset (*n* = 2153) of a population-based, cross-sectional online survey (October–December 2020), covering the “second wave” of the COVID-19 pandemic in Germany. Results: Among women, 23% increased their alcohol use, 28.4% increased their nicotine use and 44% increased their illicit substance use during the COVID-19 pandemic. Twenty percent reported major depressive symptoms and 23.4% symptoms of generalized anxiety. Generalized anxiety proved to be a significant predictor of increases in alcohol and nicotine use in logistic regression. Discussion: The mental health burden remained high during the second wave of COVID-19 and alcohol, nicotine and other substance use increased. However, the association between mental health and substance use was weak. Psychological distress does not seem to be the main motivator of substance use.

## 1. Introduction

Since the outbreak of the Coronavirus Disease 2019 (COVID-19), individuals and societies have faced various and serious health, social and economic challenges and uncertainties [1]. By early September 2021, WHO reports that there had been more than 218 million confirmed COVID-19 cases and 4.5 million related deaths globally. For Germany, the figures to date are 3.9 million confirmed COVID-19 cases and over 92,000 deaths. The economic impact of the COVID-19 pandemic cannot yet be quantified conclusively. However, there were significant declines in gross domestic product (GDP), particularly at the beginning of 2020. GDP in Germany, for example, fell by 4.9% compared with the previous year. In addition, the German government has approved 7.3 billion Euros of COVID-19-related emergency economic aid and 130 billion Euros for economic stimulus measures.

Governments have imposed widespread restrictions on public life to cope with the pandemic worldwide. The resulting “social distancing” policies and other measures to contain the incidence of COVID-19 infections led to isolation and solitude for many individuals [2]. Men are disproportionately affected by severe COVID-19 disease progression and show a higher mortality due to COVID-19 [3,4]. However, there is evidence from the scientific literature that women may be more affected by the economic, social and psychological consequences of the pandemic [5]. Sectors with a high share of female employees (e.g., hotels, restaurants, service, retail) were disproportionately affected by lockdown-related closures, short-time work and job losses [6]. Women are also more likely to work in atypical jobs with lower social security coverage or part-time, which contributes to lower financial security [7]. In the pandemic, at least in families with “traditional” gender roles, the main burden of child care at home was on women because of closed schools and child care [8]. In addition, employees in health care and other essential services (e.g., grocery retail, child or elderly care) are also mostly female [7] and have been exposed to stress, high workload levels and an increased risk of infection during the pandemic [9]. Additionally, in a student population, perceived stress with regard to COVID-19 was shown to be higher in female than in male individuals [10]. Since the beginning of the pandemic, the impact of the COVID-19 situation on mental health in the population has been discussed. At the beginning of the pandemic, researchers drew attention to the necessity to address the public mental health consequences. [11,12]. Some authors argued that a “psychiatric pandemic” was co-occurring with COVID-19 [12]. In the meantime, numerous studies in various countries showed that the COVID-19 situation has negatively affected mental health [13,14,15,16]. Population-based studies in Germany found evidence that depressive symptoms, anxiety, sleep disturbances and psychological distress increased as a response to COVID-19 [17,18,19]. Research with repeated cross-sectional designs indicated that the implementation of contact restrictions was associated with increased levels of depression and anxiety that seem to have persisted even during the easing of those restrictions [20]. Women experienced higher levels of depression and anxiety than men during the first COVID-19 wave [18,20]. This corresponds to previous research demonstrating gender differences regarding depression, anxiety and insomnia because of the pandemic [12].

Psychological distress and mental health problems are well-described risk factors for increases in the use of alcohol and other substances. Besides the potential negative effects for individuals, significant increases in alcohol use are also a public health concern. In the German general population, per capita alcohol consumption is high by global standards [21]. In the group of women in Germany, 13.1% (men: 18.5%) drink alcohol in a hazardous pattern, defined for women as more than 10 g pure alcohol per day on average [22]. Among women, alcohol consumption increases with higher socioeconomic status [22]. Women with a higher socioeconomic status are twice as likely to drink hazardously than women with a middle or low socioeconomic status [22]. A number of publications document changes in alcohol consumption under COVID-19 pandemic conditions for different countries, e.g., for the United States [23], the United Kingdom [24], Poland [25], Australia [26] and France [27]. These studies have shown that between one fifth and one quarter of adults increased their alcohol use after the pandemic started. Corresponding to those results, retailers in various countries have reported an increase in the sale of alcoholic beverages since the beginning of contact restrictions or lockdowns [28]. In contrast, a very recent study in 21 EU countries showed that in most countries, with the exception of Ireland and the United Kingdom, there was a decrease in alcohol consumption. However, a reduction in alcohol consumption was less common among people who were particularly stressed by the pandemic [29]. Other studies also found significant associations between higher levels of psychological distress, or depressive or anxiety symptoms and an increase in alcohol use. Among women, psychological distress related to COVID-19 has been significantly associated with the quantity of alcohol use, such as the number of drinks had at the last heaviest drinking event and the number of drinks on a typical occasion [23]. For women, social distancing policies and the resulting loss of social support has been associated with an increase in hazardous drinking (quantified by changes in the Alcohol Use Disorder Identification Test (AUDIT)) during the first lockdown in the USA [30]. Those increases in alcohol consumption under pandemic conditions were interpreted as a dysfunctional coping mechanism for distress caused by the pandemic [23,31,32].

The analysis refers exclusively to the female part of a population-based German sample as it is hypothesized that women have been psychologically burdened by the COVID-19 pandemic differently to men due to their professional, family and social situation. In addition, women and men are known to differ in the extent and pattern of their drinking behavior [33,34]. The objective of this analysis is to examine whether alcohol consumption changed in a German population-based sample of women under conditions of the COVID-19 pandemic. In addition, the extent of depressive symptoms, anxiety and COVID-19 specific fears and their influence on alcohol, nicotine and illicit substance consumption are analyzed.

## 2. Materials and Methods

### 2.1. Design and Sampling

A population-based and cross-sectional online survey with a self-selected convenience sample was conducted from October until December 2020. The survey covers the period of the “second wave” of the COVID-19 pandemic in Germany. During this phase, government-mandated contact restrictions to contain the pandemic in Germany were relatively strict. Schools remained closed, public life was significantly restricted and people’s daily lives changed significantly. Inclusion criteria were at least 18 years of age and German language capabilities, since the survey was only available in German. Participants were recruited via social media, institutional newsletters and online press releases. Participation was anonymous and there were no financial compensations or other incentives for participation. Electronic informed consent was obtained prior to the start of the survey. Participation was voluntary and anonymous, and participants could withdraw from the study at any time. The study was conducted in accordance with the Declaration of Helsinki, and the Ethics Committee of the University Hospitals Essen has approved the study (20-9307-BO). The Foundation of University Medicine Essen (“Stiftung Universitätsmedizin”) funded the study. The Open Access Fund of the University of Duisburg-Essen funded the publication of the study.

### 2.2. Measures

Socio-demographic data were assessed including age, gender, education, marital status, occupation, residential situation. The survey also included items on alcohol, nicotine and illicit substance use as well as gaming behavior. Mental health status was screened using PHQ 2 [35] for depressive symptoms and GAD 7 [36] for generalized anxiety. Higher scores indicate higher depressive and anxiety symptoms. Specific COVID-19-related fears were assessed by the COVID-19 Anxiety Questionnaire [37]. The higher the score, the greater the specific fear of COVID-19.

## 3. Results

### 3.1. Sample

The survey was completed by 3487 participants, the analysis refers to the subset of female participants (*n* = 2813, 80.7%). The largest age group was between 25 and 44 years (48.2%). The sample was highly educated; the majority had the highest possible school degree (32.2%) or had completed university (40.6%), 79% were employed and 63.7% lived with a partner or were married. For all sociodemographic characteristics see Table 1.

### 3.2. Mental Health and Substance Use

Generalized anxiety symptoms above the cut-off for at least moderate anxiety symptoms (>10) were found for 23.4%. Depressive symptoms above the cut-off (>3) for major depressive symptoms were reported by 20.4% of the sample. For all results of the mental health screening instruments see Table 2.

Any alcohol use in the last 30 days was reported by 67.5% of the sample; 3.5% reported drinking on more than 25 days in the last 30 days. A moderate alcohol amount with one or two drinks on each occasion was reported by 74.9% of women; binge drinking (>5 drinks per occasion) occurred in 6.8%. Over the whole sample, an increase in alcohol use was reported by 23%, whereas 18.5% reduced alcohol use. Changes in alcohol use were different between age groups. In the youngest (18–34y) and oldest (+55y) groups, changes were similar in both directions; among the younger group, 23.9% decreased and 24.4% increased their substance use; in the oldest group, 14.5% decreased and 14.5% increased their alcohol use. In the middle-aged group (35–54y), 15.3% decreased and 25.2% increased their substance use. Overall, groups differed significantly (χ^2^ = 40.063; *p* < 0.001), with a small effect size (Phi = 0.157); a post hoc test (Bonferroni-corrected *p* < 0.0125) showed that the middle-aged and oldest age groups differed significantly.

Nicotine use was reported by 23% of the sample, 28.4% smoked/vaped more and 10.9% reduced their nicotine use. Among illicit substances, cannabis was most commonly used (2.7%), followed by amphetamines (0.7%), cocaine (0.5%) and non-prescribed opioid analgesics (0.4%). Regarding illicit substance use, 44% consumed their preferred substance more often and 20% reduced their illicit substance use. Since the number of participants who reported any illicit substance use was small (*n* = 120), further analysis only refers to alcohol and nicotine use. For all substance-related results, see Table 3.

### 3.3. Alcohol

A binomial logistic regression was performed with *n* = 1898 (67.5%) participants who had reported any alcohol use in the last 30 days to determine the effect of depressive symptoms, generalized anxiety and COVID-19-related fears and predict the likelihood of an increase in alcohol use. The binomial logistic regression model was statistically significant (χ^2^ (df 3) = 99.52, *p* < 0.001), resulting in a small amount of explained variance [38] as shown by Nagelkerke’s R^2^ = 0.077.

Goodness-of-fit was assessed using the Hosmer–Lemeshow test, indicating a good model fit (χ^2^ (df 8) = 11.09, *p* = 0.196). Linearity was assessed using the Box–Tidwell procedure. All variables were found to follow a linear relationship. Bonferroni-correction (*p* = 0.13) was applied to all terms in the model [39]. The results of tolerance and variance influence factor (VIF) analysis showed that the tolerance values were all > 0.1 (PHQ 2: 0.375, GAD 7: 0.310 and Covid-19 anxiety questionnaire: 0.578) and the VIF was < 10 (PHQ 2: 2.667, GAD 7: 3.228 and Covid-19 anxiety questionnaire: 1.729) for all mental health measures, indicating that multicollinearity was not a confounding factor in the analysis. Of the three variables entered into the regression model, only anxiety (GAD 7) contributed significantly to an increase in alcohol use (*p* < 0.001). Depression (PHQ 2) and COVID-19-related fears showed no significant effect (PHQ *p* = 0.162 and Fear of Covid Scale *p* = 0.248).

### 3.4. Nicotin

A binomial logistic regression was performed with *n* = 676 participants who reported nicotine use in the last 30 days to determine the effect of depressive symptoms, generalized anxiety and COVID-19-related fears and predict the likelihood of an increase in nicotine use. The binomial logistic regression model was statistically significant (χ^2^ (df 3) = 100.706, *p* < 0.001), resulting in a small amount of explained variance [38] as shown by Nagelkerke’s R^2^ = 0.199. Goodness-of-fit was assessed using the Hosmer–Lemeshow Test, indicating a poor model fit, (χ^2^ (df 8) = 21.194, *p* = 0.007). Linearity was assessed using the Box–Tidwell procedure. All variables were found to follow a linear relationship. Bonferroni-correction (*p* = 0.013) was applied to all terms in the model [39]. The results of tolerance and variance influence factor (VIF) analysis showed that the tolerance values were all > 0.1 (PHQ 2: 0.355, GAD 7: 0.291 and Covid-19 anxiety questionnaire: 0.571) and the VIF was < 10 (PHQ 2: 2.819, GAD 7: 3.431, Covid-19 anxiety questionnaire: 1.751) for all mental health measures, indicating that multicollinearity was not a confounding factor in the analysis. Of the three variables entered into the regression model, only anxiety (GAD 7) contributed significantly to an increase in nicotine use (*p* < 0.001). Depression (PHQ 2) and COVID-19-related fears showed no significant effect (PHQ *p* = 0.557 and Fear of Covid Scale *p* = 0.154).

All model coefficients and odds are displayed in Table 4.

## 4. Discussion

In this population-based sample of women in Germany, nearly one quarter of those who use alcohol increased their alcohol use. Nearly one third of smokers increased their nicotine use and more than forty percent who used other substances increased their substance use during the COVID-19 pandemic. One in five women reported major depressive symptoms and nearly one quarter at least mild symptoms of generalized anxiety. Generalized anxiety proved to be a significant predictor of increases in alcohol and nicotine use. Depressive symptoms and specific COVID-19-related fears did not contribute significantly to an increase in alcohol or nicotine use.

### 4.1. Substance Use

The proportion of approximately one quarter of the participants who reported an increase in their alcohol corresponds with the results of previous studies [23,24,25,26], which found similar rates of alcohol use during the first wave of COVID-19. The most significant increase in alcohol consumption in the 35–55 year old group may be explained by the fact that in this group the stress of childcare, homeschooling or caring for elderly relatives may have been more prevalent. Due to the ongoing social distancing policies, the opportunity to drink alcohol at parties, bars, restaurants or events decreased markedly. This may explain the decrease in alcohol consumption in the youngest age group, in which almost the same number of women reduced as increased their alcohol consumption. Due to the social distancing measures, it is likely that the increased alcohol consumption primarily occurred in private settings with family members, closest friends or alone. Social distancing and self-isolation came along with the disruption of daily routines, boredom, loss of daily structure and lack of social contacts, which were identified as motives for a rise in alcohol consumption during the pandemic [40]. A US study also found that the longer people spent time at home, the higher the risk of binge drinking at home [41]. The relief of negative emotions and stress caused by the pandemic might have been a further motivator to drink more alcohol. Increases in alcohol consumption can negatively affect physical health in various ways; it is a leading risk factor for global disease burden and causes substantial health loss. Alcohol use is an important cause of traffic accidents and self-harm among young people and promotes various types of cancer. Alcohol use also adversely affects cardiovascular diseases such as hypertension in a dose-dependent manner. In the context of COVID-19, the negative health impacts of alcohol use are important to consider as both cardiovascular diseases and cancer increase the risk for severe COVID-19 disease progression or mortality.

With regard to nicotine, its use is also highly correlated with mental stress in women [42]. In addition, external reasons may also have led to an increase in nicotine consumption. Contact restrictions and working from home meant that many people stayed mainly in their home environment. For smokers, this may mean that the smoking bans in public and reduced social control (e.g., at work) disappeared and may have led to an increase in cigarette consumption. Active smoking is a well-studied risk factor for the development and worsening of COPD, asthma and chronic respiratory diseases. Non-smokers in households with smokers may also have been more exposed to secondhand smoke during the pandemic and associated “stay at home” policies. Passive smoking increases the risk of asthma, reduced lung function and respiratory tract infections in children. Tobacco use has a special role in the context of COVID-19 because of its negative impact on several preexisting conditions that promote the risk of severe COVID-19 disease progression (e.g., chronic respiratory or cardiovascular diseases) [43].

In the small subgroup of participants who use any illicit substances, predominantly cannabis, consumption increased considerably more than for alcohol and nicotine. These results were contrary to a Belgian sample that found no changes in cannabis use [40]. However, the results are consistent with a longitudinal Dutch study that also found an increase in cannabis use during COVID-19, but no increase in the severity of cannabis use disorder (CUD) in daily consumers [44]. In this study, mental wellbeing was reduced and contributed significantly to changes in cannabis use. In Canada, self-isolation was associated with an increase in consumption in male cannabis users; coping with depression motivated the use of more cannabis than pre-pandemic [45], and an increase in cannabis use was associated with financial concerns and lower education [46].

### 4.2. Mental Health and Substance Use

#### 4.2.1. Generalized Anxiety and Specific COVID-19 Anxiety

Symptoms of generalized anxiety were the only mental health factor that predicted an increase in alcohol and nicotine use in this sample. Previous studies have shown diverse findings regarding alcohol use and anxiety under COVID-19 conditions. An Australian study found an association between anxiety and alcohol use [26], whereas another study in the United Kingdom found no correlation between alcohol use and anxiety in an adult sample who were in self-isolation [24]. In general, the association of alcohol use and symptoms of anxiety and anxiety disorders are well documented [47]. Although the COVID-19 pandemic is a novel situation, research on other collective stressful events such as SARS 1 in 2003 [48] or the economic crisis in 2008 [49] demonstrated that those events were associated with an increase in alcohol use, partly mediated by depression and anxiety symptoms.

Evidence of specific COVID-19-related anxiety as an influencing factor is less clear. In contrast to generalized anxiety, specific fear of COVID-19 did not contribute significantly to explain changes in substance use in our study. This corresponds to findings from a US study, which also found that the subjective fear of virus infection was not associated with an increase in substance use [23]. Another international study group found, however, a significant association between COVID-19-related fear and increases in substance use among Russian, Belarusian and Israeli students during the first wave of COVID-19 [50,51].

#### 4.2.2. Depression

The finding that depressive symptoms do not significantly contribute to the increase in alcohol consumption is rather unexpected since previous studies showed the opposite effect [24,31]. In general, an increase in depressive symptoms is a risk factor for alcohol use and vice versa. Major depression and alcohol use disorder (AUD) are closely associated. The presence of either disorder doubles the risk of the second disorder [52]. Co-occurrence of AUD and depressive disorders is associated with greater severity and worse prognosis for both disorders [53]. In our female sample, one in five reported depressive symptoms above the cut-off in the PHQ, indicating that COVID-19 negatively affects mood and mental wellbeing, but drinking alcohol to cope with depression does not seem to be the preferred option.

Overall, all prediction models with mental health measures resulted in only a small amount of explained variance regarding substance use. Despite a mental burden that was heightened compared to normative samples and a subgroup of individuals who increased their alcohol and nicotine use, the association between mental health and substance use increase was not strong. Substance use to cope with anxiety and depression was, perhaps, not the main motive for increases in the use of alcohol and other substances. Characteristics of the sample may give an indication as to the causes for this lack of association. Women in this sample were highly educated, and their professions and occupational situations indicated a high socioeconomic status. The vast majority were not affected by job loss or short-time work and had not been indebted due to COVID-19, so existential concerns were unlikely. These results correspond with the findings from a large-scale, population-based study in the EU, which found that in the high-income groups an increase in alcohol use depended on the experience of financial distress. Individuals with a high income and no financial distress were more likely to decrease their alcohol use [29]. The privileged baseline of the present sample may be associated with more individual resources to cope functionally with any psychological distress that arose due to COVID-19 rather than using substances. Although alcohol consumption is generally higher in the high socioeconomic status group of women than in the low/middle socioeconomic status group, the study population did not show an excessive increase in terms of alcohol consumption.

Preexisting social inequalities in health were highlighted by the COVID-19 pandemic. Epidemiological evidence shows that morbidity and mortality risk of COVID-19 is higher in individuals with a low socioeconomic status [54,55,56]. Incidence rates were higher in districts and neighborhoods where a low socioeconomic status dominates [55,57]. This correlation might be explained by the fact that more people in these populations work in jobs that involve personal contact or mobility and cannot be done from home, or they work in jobs with low social security or are easily threatened with dismissal. This leads to more contact situations with a risk for COVID-19 transmission. In addition, some non-communicable diseases are risk factors for severe COVID-19 progression and are overrepresented in individuals with a low socioeconomic status [58,59]. These social- and health-related disadvantages played a subordinate role in this sample, which may have meant that objective and subjective threats to their own health and social situation were less pronounced in this sample.

The prevention of mental disorders and hazardous substance use due to COVID-19 are of public health relevance as both lead to a high individual and societal burden. To prevent mental distress and dysfunctional coping with alcohol or other substances, groups that are more vulnerable should be focused on. Social support has been shown to be a protective factor against increased substance use [60]. The prevention of hazardous alcohol use in the course of the current or any future collective crisis should promote social support in the community. E-mental health applications also offer a broad range of opportunities to promote mental health in times of social distancing [61]. Future e-mental health applications should also address dysfunctional substance use as an important facet of mental and physical health.

### 4.3. Limitations

The study reports on a large sample of women in the general population of Germany during the second wave of COVID-19; nevertheless, some limitations need to be considered. An online survey was used to collect the data. Thus, the possibility of selection bias needs to be considered. In addition, results may be biased in terms of socioeconomic status since high socioeconomic status was overrepresented. The sample does not reflect the average situation of women in Germany. It may be that the association between mental burden and substance would have been different if the sample had been more diverse. Due to the cross-sectional design, it is not possible to interpret mental health measures without considering that they might have been heightened before COVID-19 and that the pandemic was not causative. Moreover, mental health screening instruments do not allow the proper diagnoses of mental health disorders to be made. With regard to substance use, the quantification of substance use before COVID-19 was limited, so it remains unclear as to what extent substance use increased.

## 5. Conclusions

The psychological consequences of COVID-19 seem to have affected different segments of the population to different degrees, just as the infection itself does. Despite heightened symptoms of depression and anxiety in this sample, higher personal and economic resources possibly allowed individuals to cope with increased stress, depression and anxiety in a more functional way. An exception to this appears to be the subgroup of illicit substance users, who increased the use of their preferred addictive substance almost twice as much as that of alcohol and nicotine users.

## Figures and Tables

**Table 1 ijerph-18-09728-t001:** Sociodemographic data.

	*n*	%
Age		
18–24	439	15.6
25–34	731	26
35–44	624	22.2
45–54	552	19.2
55–64	348	12.4
>65	119	4.3
Education		
university degree	1143	40.6
university attendance certificate	909	32.3
secondary education	737	26.2
no school certificate	9	0.3
others	15	0.5
Relationship status		
married/partner	1792	63.7
single	770	27.4
divorced	192	6.8
widowed	38	1.4
others	21	0.7
Children under 18 years	789	28
Occupational status		
employee	1704	79
freelancer	194	9
civil service	133	4.7
others	126	5.8
Unemployment	76	2.7
Changes occupational/financial situation COVID-19		
job threatened	398	14.1
short-time work	304	10.8
take debts	156	5.5
job loss	143	5.1
Preexisting chronic conditions		
mental disorders	687	24.4
chronic respiratory diseases	348	12.4
hypertension	319	11.3
diabetes	90	3.2
cardiovascular diseases	81	2.9

**Table 2 ijerph-18-09728-t002:** Descriptive data mental health measures (*n* = 2813).

	M	MED	SD	Min	Max	>Cut-Off *
Depressive Symptoms (PHQ 2)	1.9	2	1.8	0	6	20.4%
Generalized Anxiety (GAD 7)	6.8	6	5.5	0	21	23.6%
COVID-19-Anxiety Questionnaire	19.1	18	6.9	1	40	-

* PHQ-2 > 3 major depressive symptoms; GAD 7: >10 at least moderate generalized anxiety).

**Table 3 ijerph-18-09728-t003:** Description substance use under pandemic conditions.

	*n*	%
Any alcohol use last 30 days	1898	67.5
Days with alcohol use (last 30 days)		
no alcohol use	915	32.5
1–8	1308	46.5
9–24	491	17.5
25 and more	99	3.5
Drinks */occasion		
1–2	1422	74.9
3–4	346	18.2
5–6	85	4.5
7–8	37	1.9
9 or more	8	0.4
Changes alcohol use (*n* = 1898)		
less alcohol use	351	18.5
no difference	1111	58.5
more alcohol use	436	23
Nicotine use	676	24
Changes nicotine use (*n* = 676)		
less nicotine use	74	10.9
no difference	410	60.7
more nicotine use	192	28.4
Other substance use		
Cannabis	75	2.7
Amphetamine	20	0.7
Cocaine	13	0.5
Opioid analgesics/sedatives (not prescribed)	12	0.4
Changes other substance use (*n* = 120)		
less substance use	24	20
no difference	43	35.8
more substance use	53	44.2

* definition one drink: 0.33 L beer; 0.25 L wine; 0.02 L spirituous beverage.

**Table 4 ijerph-18-09728-t004:** Model coefficients binominal logistic regression changes in alcohol and nicotine use.

	B	SE	Wald	df	*p*	Odds Ratio Exp (B)	95% CI	
Alcohol								
PHQ 2	0.067	0.048	1.957	1	0.162	1.069	0.973	1.175
GAD 7	0.074	0.018	17.076	1	0.000	1.077	1.040	1.116
COVID-19 Anxiety Questionnaire	0.012	0.0110.	1.135	1	0.248	1.013	0.991	1.034
Constant	−2.094	0.176	141.745	1	0.000	0.123		
Nicotine								
PHQ 2	0.044	0.075	0.345	1	0.557	1.045	0.902	1.211
GAD 7	0.121	0.028	18.371	1	0.000	1.129	1.068	1.194
COVID-19 Anxiety Questionnaire	0.022	0.016	2.029	1	0.154	1.023	0.992	1.054
Constant	−2.463	0.280	77.398	1	0.000	0.085		

## Data Availability

The data presented in this study are available on request from the corresponding author.

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
