# Peer review of "Changes in Substance Use and Mental Health Burden among Women during the Second Wave of COVID-19 in Germany"

_ijerph, 2021, doi:10.3390/ijerph18189728_

Round 1
Reviewer 1 Report
1.- In the abstrect part, delete the term introduction, it is understood that you have to introduce the subject.
2.-the abstract should be better written for better understanding of the reader.
3.- add updated figures of COVID cases in the world and in Germany, at the beginning of the introduction.
4.- Add information on economic losses, in introduction.
5.- partly experimental, it is necessary to clarify which variables were those that were measured.
6.- It is considered that the following paragraph should be partly experimental:
The survey was completed by 3487 participants, the analysis refers to the subset of female participants (N = 2813, 80.7%). The largest age group was between 25 and 44 years (48.2%). The sample was highly educated; the majority had the highest possible school degree (32.2%) or completed university (40.6%), 97.3% were employed and 63.7% lived with a partner or were married. For all sociodemographic characteristics see table 1.
7.- the tables cannot be seen in the document.
8.-Missing adding figures or tables that illustrate the results in the text.
9.- improve the discussion of results significantly.
Author Response
Please see attachment for responses to your review.

Reviewer 2 Report
This study aims to understand the substance use and mental health burden among women during the second wave of COVID-19 in Germany, by using a population based and cross-sectional online-survey with a self-selected convenience sample during October-December 2020. The study is interesting, since most of the participants were high SES and employed.
- The authors’ defined freelancer, civil service, and others all as employed. Although they are self-employed or employed by the government, the nature of the employment is different especially during pandemic.
- Table 2 was not mentioned in the main text at all.
- If would be nice if the authors could stratify nicotine use to different products – cigarette, smokeless tobacco, or vaping, so we can understand the differences in increase and decrease in different product use.
- Changes in substance use is likely a change in health behavior – does the author look at the alcohol users and identify the changes in their nicotine and illicit substance use?
- Did the authors mean that multicollinearity was not a confounding factor for the following variables: the effect of depressive symptoms, generalized anxiety, and COVID-19 related fears for both models (alcohol and nicotine)? Multicollinearity should be determined by examining tolerance and the Variance Inflation Factor (VIF) and not by the correlations between predictor variables. They are so interconnected that I would expect some collinearity.
Author Response

(The authors gave the same response as above.)

Round 2
Reviewer 1 Report
Add more references in the results and discussion part to improve the content of the article, mainly on how alcohol and nicotine affect behavior, and what consequences these abuses have on physical health.
Author Response
Thank you for the clarification! I have added references to the possible negative physical effects of increased alcohol use starting at line 235, and starting at line 248 you will find additions to the influence of smoking. At the same time, I have tried to place these factors in the context of COVID-19. Since the article is already very long and close to the word limit of the journal, I refrained from including additional factors.
However, please do not hesitate to provide me with renewed feedback if you find further additions useful.